# Cell-Surface Programmed Death Ligand-1 Expression Identifies a Sub-Population of Distal Epithelial Cells Enriched in Idiopathic Pulmonary Fibrosis

**DOI:** 10.3390/cells11101593

**Published:** 2022-05-10

**Authors:** Negah Ahmadvand, Gianni Carraro, Matthew R. Jones, Irina Shalashova, Afshin Noori, Jochen Wilhelm, Nelli Baal, Farhad Khosravi, Chengshui Chen, Jin-San Zhang, Clemens Ruppert, Andreas Guenther, Roxana M. Wasnick, Saverio Bellusci

**Affiliations:** 1Cardio-Pulmonary Institute, Department of Pulmonary and Critical Care Medicine and Infectious Diseases, Universities of Giessen and Marburg Lung Center (UGMLC), Member of the German Center for Lung Research (DZL), Justus-Liebig University Giessen, 35392 Giessen, Germany; negah.ahmadvand@innere.med.uni-giessen.de (N.A.); matthew.jones@innere.med.uni-giessen.de (M.R.J.); irn.bry@yandex.ru (I.S.); afshin.noori@med.uni-giessen.de (A.N.); clemens.ruppert@innere.med.uni-giessen.de (C.R.); andreas.guenther@innere.med.uni-giessen.de (A.G.); 2Lung and Regenerative Medicine Institutes, Cedars-Sinai Medical Center, Department of Medicine, Los Angeles, CA 90048, USA; gianni.carraro@csmc.edu; 3Institute for Lung Health (ILH), Department of Internal Medicine, Justus-Liebig University Giessen, 35392 Giessen, Germany; jochen.wilhelm@patho.med.uni-giessen.de; 4Institute for Clinical Immunology and Transfusion Medicine, Justus-Liebig University Giessen, 35392 Giessen, Germany; nelli.baal@immunologie.med.uni-giessen.de; 5Department of Physiology, Justus-Liebig University Giessen, 35392 Giessen, Germany; farhad.khosravi@physiologie.med.uni-giessen.de; 6The Quzhou Affiliated Hospital of Wenzhou Medical University, Quzhou People’s Hospital, Quzhou 324000, China; chenchengshui@wmu.edu.cn (C.C.); zhang_jinsan@163.com (J.-S.Z.); 7European IPF Registry/UGLMC Giessen Biobank, 35392 Giessen, Germany; 8Lung Clinic Waldhof-Elgershausen, 35753 Greifenstein, Germany; 9Laboratory of Extracellular Lung Matrix Remodelling, Department of Internal Medicine, Universities of Giessen and Marburg Lung Center (UGMLC), Member of the German Center for Lung Research (DZL), Justus-Liebig University Giessen, 35392 Giessen, Germany

**Keywords:** CD274, PD-L1, IPF, IAAP, lung progenitor, immune privilege

## Abstract

Idiopathic lung fibrosis (IPF) is a fatal lung disease characterized by chronic epithelial injury and exhausted repair capacity of the alveolar compartment, associated with the expansion of cells with intermediate alveolar epithelial cell (AT2) characteristics. Using *Sftpc^CreERT2/+^*: *tdTomato^flox/flox^* mice, we previously identified a lung population of quiescent injury-activated alveolar epithelial progenitors (IAAPs), marked by low expression of the AT2 lineage trace marker tdTomato (Tom^low^) and characterized by high levels of Pd-l1 (Cd274) expression. This led us to hypothesize that a population with similar properties exists in the human lung. To that end, we used flow cytometry to characterize the CD274 cell-surface expression in lung epithelial cells isolated from donor and end-stage IPF lungs. The identity and functional behavior of these cells were further characterized by qPCR analysis, in vitro organoid formation, and ex vivo precision-cut lung slices (PCLSs). Our analysis led to the identification of a population of CD274^pos^ cells expressing intermediate levels of *SFTPC*, which was expanded in IPF lungs. While donor CD274^pos^ cells initiated clone formation, they did not expand significantly in 3D organoids in AT2-supportive conditions. However, an increased number of CD274^pos^ cells was found in cultured PCLS. In conclusion, we demonstrate that, similar to IAAPs in the mouse lung, a population of CD274-expressing cells exists in the normal human lung, and this population is expanded in the IPF lung and in an ex vivo PCLS assay, suggestive of progenitor cell behavior. CD274 function in these cells as a checkpoint inhibitor may be crucial for their progenitor function, suggesting that CD274 inhibition, unless specifically targeted, might further injure the already precarious lung epithelial compartment in IPF.

## 1. Introduction

Alveolar epithelial type 2 cells (AT2s) are surfactant-producing cells that serve as alveolar progenitors and play essential roles in the innate immune response. AT2s interact with macrophages through the secretion of various cytokines in response to pathogens and alveolar damage. As a result, AT2s activate macrophages and defend the alveolus [1,2]. Furthermore, AT2 interaction with Foxp3^pos^ Treg cells is critical for the repair of the epithelium during lung regeneration. Additionally, AT2s display enhanced proliferation when co-cultured with Foxp3^pos^ Treg cells [3]. However, our knowledge about the role of the interaction of AT2s with different immune cells in the repair of the epithelium or during the pathogenesis of lung diseases remains limited [1,2,4,5].

AT2 heterogeneity has been demonstrated in different mouse models during homeostasis and regeneration/repair [5,6,7,8]; however, the presence and function of the human equivalent of mouse AT2 progenitor cells are largely unknown [9,10]. Using *Sftpc^CreERT2/+^*; *tdTomato^flox/flox^* mice, and based on the differential level of the tomato reporter between two lineage-traced alveolar epithelial subpopulations, we have previously reported the existence of a novel population of AT2s, named Tom^low^ AT2s, that are enriched in programmed death-ligand1 (Pd-l1) [11]. This AT2 subpopulation expresses low levels of *Sftpc*, *Sftpb*, and *Sftpa1*, and displays a low level of Fgf signaling activation. Following pneumonectomy, Tom^low^ AT2s are activated and display progenitor-like properties, as they are amplified and exhibit elevated expression of Fgf signaling genes *Fgfr2b*, *Etv5*, AT2 differentiation marker *Sftpc*, and cell cycle genes *Ccnd1*, *Ccnd2* expression, compared with sham. These changes are, therefore, consistent with the activation of the progenitor-like properties of the Tom^low^, which allows them to proliferate and differentiate toward mature AT2s.

In addition, analysis of the behavior of these cells ex vivo in precision-cut lung slices (PCLSs) from *Sftpc^CreERT2/+^*: *tdTomato^flox/flox^* mice supports their progenitor-like function in the context of significant injury to the AT2 lineage. Therefore, we named these Tom^low^ AT2s “injury-activated alveolar progenitors” (IAAPs) [12]. Pd-l1 is an immune inhibitory membrane receptor–ligand expressed in different immune and epithelial cells. Pd-l1 ligands bind to programmed death-1 (Pd-1) receptors. Pd-1 is expressed on a subset of Cd4^pos^ and Cd8^pos^ T cells, natural killer T cells, B cells, and activated monocytes. Pd-l1 is an immune inhibitory molecule that controls the inflammatory response to injury [13,14,15,16,17]. Remarkably, human PD-L1 expression is upregulated in non-small-cell lung cancer and adenocarcinoma. In the tumor microenvironment, cancer epithelial stem cells utilize the PD-L1 pathway to escape immune system surveillance by suppressing the cytotoxic response following the binding of PD-L1 ligands expressed by cancer stem epithelial cells to the PD-1 receptor on T cells [18,19,20,21].

Interestingly, the presence of membrane PD-L1^pos^ alveolar and bronchiolar epithelial cells in a subgroup of IPF patients was previously reported. The presence of these cells is associated with higher fibroblast foci detection and patchy fibrosis, compared with patients negative for PD-L1. However, no correlation between the expression of PD-L1 and honeycomb formation could be detected [22]. We also found in the normal human lung a subcluster of AT2 *PD-L1*^high^ cells following data mining of the scRNA-seq database [10]. These AT2 *PD-L1*^high^ cells express low levels of *ETV5*, *SFTPC*, and *SFTPA1* and are enriched in the expression of immune system-related genes (*CXCL1*, *CXCL2*, *CXCL3*, *CXCL8*, *ICAM1*, and *IRF1*) [11]. Intriguingly, an increased number of AT2 *PD-L1*^high^ cells expressing chemokines was also observed in the epithelium of patients with chronic obstructive pulmonary disease. These cells display higher expression of inflammation-related genes such as *CXCL1*, *CXCL8*, *CCL2*, and *CCL20*, which supports a strong interaction of these cells with immune cells [23].

In this paper, we analyzed the available scRNA-seq dataset in the IPF cell atlas to investigate the presence of equivalent IAAP PD-L1^pos^ cells in human lungs. In particular, we quantified by flow cytometry the abundance of EpCAM^pos^HTII-280^neg^PD-L1^pos^ cells (potentially equivalent to the IAAPs), as well as EpCAM^pos^HTII-280^pos^PD-L1^neg^ cells (equivalent to mature AT2s) in IPF vs. donor human lungs. We also examined the expression of key genes in these populations by qPCR and monitored the growth of these cells in organoid and precision-cut lung slice assays. Our data indicate that the equivalent of IAAPs, initially identified in mice, exist in humans, and are differentially regulated in IPF vs. donor lungs.

## 2. Materials and Methods

### 2.1. Human Specimens

Human lung tissues from idiopathic pulmonary fibrosis (IPF) patients undergoing lung transplantation and non-IPF donors were collected from the European IPF Registry (euIPFreg) and provided by the UGMLC Giessen Biobank, a member of the DZL platform biobanking. The study protocol was approved by the Ethics Committee of the Justus-Liebig University School of Medicine (No. 111/08: European IPF Registry and 58/15: UGMLC Giessen Biobank), and informed consent was obtained in written form from each subject. Explanted lungs (n = 9 for sporadic IPF) or non-utilized donor lungs or lobes fulfilling transplantation criteria (n = 12; human donors) were obtained from the Department of Thoracic Surgery in Giessen, Germany, and Vienna, Austria. All IPF diagnoses were made according to the American Thoracic Society (ATS)/European Respiratory Society (ERS) consensus criteria (American Thoracic Society Idiopathic Pulmonary Fibrosis: Diagnosis and Treatment International Consensus Statement, n.d. 343434), and a usual interstitial pneumonia (UIP) pattern was proven in all IPF patients.

### 2.2. Mouse Experiments

Animal studies were performed in accordance with the Helsinki convention for the use and care of animals and were approved by the local authorities at Regierungspräsidium Giessen (G7/2017-No. 844-GP and G11/2019-No. 931-GP).

### 2.3. Human Lung Tissue Dissociation

Subpleural lung tissue from explanted IPF lungs or excised from donor lungs, due to donor–recipient size mismatch, was dissected and placed on ice. Tissue was minced with three scissors to pieces of approximately 1–2 mm in diameter, washed three times with RPMI, and incubated for 30 min at 37 °C in RPMI containing dispase (BD Corning, 1:10 dilution) and 30 µg/µL DNase I, with continuous and gentle agitation on a rotating shaker. Tissue pieces were first strained using surgical gauze and then with 100 µm, 70 µm, and 40 µm cell strainers, and the strained solution was centrifuged and resuspended in RPMI containing 10% FBS. Cells were washed two more times with RPMI 10% FBS, counted, and shock-frozen in liquid nitrogen at a concentration of 2–10 × 10^6^ per vial until further use.

### 2.4. Mouse Lung Tissue Dissociation

Saline perfused lungs were removed from the thoracic cavity, and 1ml RPMI solution containing 10% dispase and 30 µg/µL DNase I was instilled intratracheally and placed in a conical tube with 3 mL of the same dissociation solution. Following 30 min incubation in a 37 °C water bath, the lung lobes were isolated, minced, and resuspended in 10 mL RPMI 10% FBS 30 µg/µL DNase I. The lung suspension was then pipetted repeatedly up and down until the individual lung pieces consolidated in one single piece of floating extracellular material and the solution became turbulent. The resulting cell suspension was filtered through 70 μm and 40 μm cell strainers. Cells were then spun down two times at 1200 RPM for 5 min, resuspended in FACS buffer (HBSS 2% FBS 10 µg/mL DNase I, 2 mM HEPES, 1% penicillin–streptomycin), and placed on ice in preparation for flow cytometry staining. 

### 2.5. Flow Cytometry and Sorting 

All analyses were performed on Li–nitrogen stored cell preparation available through the European IPF Registry biobank. Human lung cell suspensions were thawed rapidly, and cells were washed in 10 times the freezing volume of RPMI 10% FBS and maintained on ice throughout the procedure. Cells were resuspended in FACS buffer (HBSS, 2% FBS, 2 μg/μL DNase I, 2 mM HEPES, 1% penicillin–streptomycin) at a concentration of 10^6^ cells/100 μL, and cell-surface staining was performed using antibody master mixes whenever possible (antibodies in Appendix A). For intracellular staining, cell-surface staining was first performed, and then cells were fixed and permeabilized for 30 min at room temperature using the Foxp3/Transcription Factor Staining Buffer Set (eBiosciences, San Diego, CA, USA), according to the manufacturer’s instructions. Cells were washed following fixation and resuspended in 100 μL of permeabilization/wash buffer containing the primary antibody and incubated for one hour at room temperature, washed three times, followed by Anti-rabbit IgG (H + L), F (ab’)_2_ fragment (Alexa Fluor^®^ 555 Conjugate Cell Signaling Technologies, Danvers, MA, USA 1:1000).

The following controls were used: single-color controls for instrument set-up, fluorescence-minus-one (FMO—a sample where one fluorophore was omitted) controls for gating, and no primary controls for indirect intracellular staining. 

Data were acquired on a BD FACSCanto II (BD Biosciences, Franklin Lakes, NJ, USA) using BD FACSDiva software (BD Biosciences, Franklin Lakes, NJ, USA). Data were further analyzed using FlowJo versionX software (FlowJo, LLC, Ashland, Oregon).

For cell-sorting experiments, samples were thawed successively and stained as above. Cells were sorted through a 100 µm nozzle in RPMI 10%, and the sorting purity was assessed for each sample. Cells were immediately centrifuged and resuspended in lysis buffer (RNA Mini kit) for RNA isolation. The entire procedure was performed at 4 °C, and cell lysates were stored at −80 °C until RNA isolation was performed. 

### 2.6. RNA Extraction and Quantitative Real-Time PCR 

Following lysis of FACS-isolated cells from human and mouse lungs in RLT plus, RNA was extracted using a RNeasy Plus Micro Kit (QIAGEN, Venlo, The Netherlands), and cDNA synthesis was carried out using a QuantiTect Reverse Transcription Kit (QIAGEN, Venlo, The Netherlands), according to the instructions provided by the supplier. Thereafter, selected primers were designed using NCBI’s Primer-BLAST option (https://www.ncbi.nlm.nih.gov/tools/primer-blast/ (accessed on 15 September 2020); primer sequences in Appendix A). Quantitative real-time polymerase chain reaction (qPCR) was performed using a PowerUp SYBR Green Master Mix Kit, according to the manufacturer’s protocol (Applied Biosystems, Bedford, MA, USA) and a LightCycler 480 II machine (Roche Applied Science, Basel, Switzerland). *Hypoxanthine–guanine phosphoribosyltransferase* (*Hprt*) and *Glyceraldehyde 3–phosphate dehydrogenase* (*GAPDH*) were used as reference genes for mice and humans, respectively. Data were presented as mean expressions relative to *Hprt* and *GAPDH*. Data were assembled using GraphPad Prism software (GraphPad Software, La Jolla, CA, USA). Statistical analyses were performed utilizing two tailed-paired Student’s *t*-tests. Results were considered significant when *p* < 0.05.

### 2.7. Microarray Data

Differential gene expression was investigated using microarray analysis. Depending on the amount of RNA isolated per sample in an experiment, one of two possible microarray protocols was used. For RNA concentrations >50 ng/μL, the T7-protocol was followed. In this protocol, purified total RNA was amplified and Cy3-labeled using a LIRAK kit (Agilent Technologies, Waldbronn, Germany), following the kit instructions. Per reaction, 200 ng of total RNA was used. The Cy3-labeled DNA was hybridized overnight to 8 × 60 K 60 mer oligonucleotide spotted microarray slides (Agilent Technologies, Waldbronn, Germany, design ID 028005). For experiments in which samples yielded <50 ng/μL of RNA, the SPIA protocol was utilized. In this protocol, purified total RNA was amplified using an Ovation PicoSL WTA System V2 Kit (NuGEN Technologies, Leek, The Netherlands). Per sample, 2 μg amplified cDNA was Cy3-labeled using a SureTag DNA labeling Kit (Agilent Technologies, Waldbronn, Germany). The Cy3-labeled DNA was hybridized overnight to 8 × 60 K 60 mer oligonucleotide spotted microarray slides (Agilent Technologies, Waldbronn, Germany, design ID 074809). For each protocol, hybridization and subsequent washing and drying of the slides were performed following the Agilent hybridization protocol. The dried slides were scanned at 2 μm/pixel resolution using the InnoScan is 900 (Innopsys, Carbonne, France). Image analysis was performed with Mapix 6.5.0 software, and calculated values for all spots were saved as GenePix results files. Stored data were evaluated using the R software (version 3.3.2) and the limma package (version 3.30.13) from BioConductor (https://www.bioconductor.org, accessed on 20 October 2020). Gene annotation was supplemented by NCBI gene IDs via biomaRt.

### 2.8. Organoid Assay

Sorted AT2 (CD45^neg^ CD31^neg^ EpCAM^pos^ HTII-280^pos^) and CD274^pos^ (CD45^neg^ CD31^neg^ EpCAM^pos^ HTII-280^neg^ CD274^pos^) cells from both donor and IPF lungs were centrifuged and resuspended in cell culture medium (RPMI Life Technologies, 10% FBS). Donor interstitial fibroblasts from one donor lung were provided by the Giessen Biobank [24] and were used as the mesenchymal component of the organoids. Mesenchymal (4000 cells/insert) and epithelial cells (5000 AT2s or 4000 CD274^pos^) suspensions were mixed in 100 μL cell culture media, followed by the addition of cold Matrigel^®^ growth factor-reduced (Corning) at a 1:1 dilution resulting in 200 μL final volume per insert. The Matrigel^®^ cell suspension was placed on the top of the filter membrane of the insert and incubated at 37 °C for 30 min. Next, 350 μL of the medium was transferred to each well. Cells were incubated under air–liquid conditions at 37 °C with 5% CO_2_ for 12 days. Media were changed three times per week.

### 2.9. PCLS Culture and Dissociation

One segment of each explanted human IPF lung was filled with 1.5% Low Melt Agarose (Bio-Rad) at 37 °C and allowed to cool on ice for 30 min. Blocks of tissue of ~3 cm × 3 cm × 4 cm (depth × width × height) were cut and prepared for sectioning using a Vibrating Blade Microtome (Thermo Fisher, Waltham, MA, USA) at a thickness of 500 μm. PCLSs were cultured for 3 days in DMEM/F12 (Gibco, Thermo Fisher, Waltham, MA, USA), supplemented with 10% human serum (Human Serum Off The Clot, Biochrom, Berlin, Germany) and 1% penicillin–streptomycin in the presence of DAPT (10–50 μM in DMSO) or DMSO (1:1000; Sigma-Aldrich, Burlington, MA, USA). The medium was changed daily. At the end of the experiment, PCLSs were dissociated using Dispase (1:10 dilution; Roche, Basel, Switzerland) and 10 μg/μL DNase I for 45 min at 37 °C and processed for flow cytometry (see above). 

## 3. Results

### 3.1. CD274 mRNA Expression in Donor and IPF Lungs

We previously identified a population of AT2s in mice and humans that express high levels of CD274 (also known as PD-L1) on their surface. In mice, these cells have quiescent progenitor characteristics. In humans, their role during homeostasis and in pathological conditions remains to be elucidated. To determine the mRNA expression level in the epithelial compartment at the single-cell level, two recently published scNGS datasets of donor and IPF lung cells (Kaminski and Banovich) were investigated (Appendix A). In both datasets, *CD274* expression was found in a subpopulation of AT2s, as well as in IPF-specific aberrant basaloid and KRT5-/KRT17+ cells (Appendix A). Although the expression pattern was concordant between the two datasets, they disagreed on the change in *CD274* expression level between the donor and IPF AT2s (Appendix A). To determine the potential functional role of CD274 in pathogenesis, we searched the same databases for the expression of its cognate receptor *PD-1* (also known as the *PDCD1* gene) and found that it is specifically expressed in T cells, and its expression was increased in IPF (Appendix A). Thus, the data demonstrate that *CD274* is expressed in human alveolar epithelial cells, and its expression is modulated in IPF. The complementary expression of its receptor on the surface of T cells suggests CD274 may play a functional role in the alveolar epithelium-immune cell interaction. This result is supported by the previously reported high-level expression of immune-related genes in IAAPs [11].

Next, focusing on the alveolar epithelial lineage (Figure 1A), we mined existing scRNAseq (Kaminski [25], Kropski [26], and Stripp [27]; Figure 1B) for the presence of *CD274*^pos^ cells in donor lungs (Figure 1C), as well as in pathological lungs belonging to the interstitial lung diseases (ILDs) spectrum, including IPF, NSIP, unclassifiable ILD, cHP, and Sarcoidosis (Figure 1D–H). Our results indicated that *CD274*^pos^ cells were detected in the alveolar lineage of donor and pathological lungs (Figure 1I). A comparison of *CD274*^pos^ cells in control vs. fibrotic cells (across all ILDs) indicated that *CD274* expression is significantly increased in fibrotic vs. control lungs (*p* = 2.2 × 10^−16^) (Figure 1J). This difference was also significant for the other aforementioned diseases (Figure 1K), suggesting that *CD274* expression is increased in ILDs. In the mouse model, we also reported that *Fgfr2b*, *Etv5*, and the differentiation markers *Sftpa*, *Sftpc*, and *Sftpb* were upregulated in activated IAAPs following pneumonectomy [11]. This result suggested that, in mice, there is a general activation of Fgfr2b signaling in IAAPs in the context of repair. This prompted us to investigate the status in human PD-L1^pos^ cells of the Fgfr2b signature identified in the mouse at embryonic day (E) 12.5 [28], E14.5 [29], and E16.5 [30]. In the adult lung, this Fgfr2b signature is mostly present in the alveolar epithelium and is found to be conserved between mice and humans ([30] and data not shown). Our results indicated that this signature significantly increased in *CD274*^pos^ cells in IPF vs. control (Figure 1L–N), suggesting the reactivation of this AT2-specific developmental pathway in this population.

However, and as a note of caution, a direct comparison of two key genes in the FGFR2b signature in humans in the donor (n = 4) and IPF (n = 4) *CD274*^pos^ populations isolated for this study from our biobank revealed that IPF *CD274*^pos^ cells expressed lower levels of *FGFR2b* and *ETV5*, compared with the corresponding expression in donor CD274^pos^ cells (Appendix A); this suggests important heterogeneity in terms of FGFR2b signaling activation in IPF vs. donor *CD274*^pos^ cells between different datasets. The significance of FGFR2b signaling in the *CD274*^pos^ cells is still unclear and may be related to the capacity of these cells to display some of their progenitor-like characteristics, such as proliferation and/or differentiation. 

### 3.2. Cell-Surface CD274 Expression in Donor and IPF Lungs

To determine the functional pool of CD274, five donor and five end-stage IPF lung samples were dissociated into single-cell suspension and analyzed by flow cytometry. CD274 cell-surface expression was analyzed in the epithelial compartment (CD45^neg^ CD31^neg^ EpCAM^pos^) as a function of HTII-280, an AT2 specific marker (Appendix A). Confirming previously published observations [31], the proportion of AT2s (Figure 2A, population Q1) was drastically diminished in IPF—from an average of 80% in donors to approximately 25% in IPF epithelium. CD274 expression was not detected in AT2s, but a small HTII-280^neg^ CD274^pos^ population (called CD274^pos^; from here on, population Q3) was present in the donor and was statistically increased in the lungs of IPF patients (Figure 2A,B, Appendix A, population Q3 donor vs. IPF = 0.54% ± 0.255% vs. 10.48% ± 10.32%, respectively, *p* < 0.001). Notably, the amount of cell-surface CD274 was similar between donor and IPF cells, shown by the similar mean fluorescence intensity (MFI) of the Q3 population between the donor and IPF (Figure 2C, average Log (MFI) donor vs. IPF = 3.81 ± 0.08 vs. 3.92 ± 0.18, *p* = 0.18). 

In the mouse lung, CD274^pos^ cells expressed low levels of the AT2 marker proSP-C, and lineage tracing experiments showed that they belong to the AT2 lineage. Thus, we analyzed the proSP-C and CD274 expression in the epithelial compartment of the same donor and IPF samples, as in the previous analysis. Similar to the HTII-280 data, the number of proSP-C^pos^ cells was drastically decreased (Q5, donor vs. IPF = 81.9% ± 7.05% vs. 23.9%± 25.1%, *p* = 0.014, Figure 2D,E and Appendix A), and the number of CD274^pos^ cells was increased in the IPF lung (Q7 donor vs. IPF = 0.84 ± 0.34 vs. 13.58 ± 11.2, *p* < 0.001). We also found that the CD274^pos^ population was entirely proSP-C-negative (Figure 2D,E). Taken together, these data suggest that, in the human lung, CD274 expression is confined to a population of HTII-280^neg^ proSP-C^neg^ epithelial cells that are increased in the IPF lung. 

### 3.3. Molecular Characterization of the CD274^pos^ Population in Mouse and Human Lungs

To gain insight into the identity of the corresponding CD274^pos^ population in the human lung, four donor and four IPF epithelial populations were sorted according to their HTII-280 vs. CD274 expression pattern (Figure 3A, populations Q1—HTII-280^pos^ CD274^neg^, Q3—HTII-280^pos^ CD274^pos^ and Q4—HTII-280^neg^ CD274^neg^), and the expressions of the *SFTPC* and *SCGB1A1* lineage-specific markers were determined in these isolated subpopulations by qPCR (Figure 3B). The Q1 and Q4 populations had patterns of expression of *SFTPC* and *SCGB1A1* consistent with their respective alveolar and conducting airway lineages, while the Q3 population (CD274^pos^) cells expressed intermediate levels of *SFTPC*. The difference in the expression levels of *SCGB1A1* between Q3 and Q1 subpopulations in either donor or IPF was not statistically significant (Figure 3B,C). 

Our previously published data [11] led to the identification of a population of alveolar epithelial cells that express low levels of the lineage tracing fluorescence protein tdTomato (Tom^low^) alongside mature AT2s (Tom^high^, Figure 3C). The Tom^low^ population is enriched in Cd274^pos^ cells and bears quiescent progenitor capabilities [11]. To understand the molecular phenotype of these populations, we performed a qPCR analysis of *Sftpc* (an AT2 marker) and *Scgb1a1* (a club cell marker) on sorted Tom^high^ (corresponding to the Q1 subpopulation), Tom^low^ (corresponding to the Q3 subpopulation). and Tom^neg^ cells (corresponding to the Q4 subpopulation) from tamoxifen-treated *Sftpc^CreERT2/+^*: *tdTomato^flox/flox^* mice. As expected, the Tom^high^ cells expressed high levels of the AT2 marker *Sftpc* and minimal levels of the club cell marker *Scgb1a1*, while the Tom^neg^ cells had the opposite expression pattern, conforming to their conducting airway identity. Interestingly, the Tom^low^ population had a level of *Sftpc* expression lower than mature AT2s but significantly higher than Tom^neg^, supporting our conclusion that Tom^low^ cells belong to the AT2 lineage (Figure 3C). Interestingly, Tom^Low^ cells displayed low levels of expression of *Scgb1a1*, similar to what was observed in Tom^high^ cells (Figure 3C). 

Next, we carried out bulk RNA sequencing of isolated AT2s (Q1) and PD-L1 (Q3) subpopulations from n = 2 donor and n = 2 IPF patients. First, we explored the status of the AT2 signature in AT2s vs. PD-L1^pos^ cells in donor and IPF. Similar to what we found in the mouse [11], we found significantly lower levels of AT2 markers in PD-L1^pos^ cells vs. AT2s (Appendix A). Interestingly, in our limited dataset, we did not observe any significant difference in the AT2 signature within PD-L1^pos^ cells of either donor or IPF. This result supports our previous observation that FGFR2b signaling, which is known to promote AT2 differentiation, is not increased in PD-L1^pos^ cells in IPF (Appendix A)

Next, we identified the most regulated pathways in AT2s and PD-L1^pos^ cells in IPF vs. donor (Appendix A). In AT2s, the most dysregulated pathways were ribosome, focal adhesion, staphylococcus infection, and chemokine signaling pathways. In PD-L1^pos^ cells, a very significant increase in the dynamic range of the changes observed between IPF and donor was noted, compared with the dynamic range of changes observed in AT2s (−log_10_ (*p*-value)) = 23 vs. 12 in PD-L1^pos^ cells vs. AT2s, respectively). This result suggests tonic transcriptomic changes in the PD-L1^pos^ cells between donors and IPF. The most relevant dysregulated pathways in PD-L1^pos^ cells were pathways in cancer, cell cycle, and Hippo signaling. This limited transcriptomic analysis, which should be expanded in the future, suggests that PD-L1^pos^ cells display a lower AT2 signature, compared with mature AT2s, and exhibit dysregulation in the expression of genes linked to cell proliferation.

### 3.4. In Vitro Clonogenic Potential

Mouse Tom^low^ cells combined with resident mesenchymal cells gave rise to fewer organoids in 3D culture, compared with the AT2 population of Tom^high^ cells [11]. To study the in vitro clonogenic potential of the CD274^pos^ population, CD274^pos^ (Q3 population) and AT2s (Q1 population) were sorted from two donor patients and cultured in Matrigel^®^ in the presence of donor-derived interstitial fibroblasts and in a medium permissive of AT2 growth for 12 days (see Materials and Methods). In this assay, sorted AT2s gave rise to alveolospheres at the expected frequency (3–5%, Figure 4A). Although organoid formation was initiated in the CD274^pos^ population, the organoids failed to grow in the provided conditions (Figure 4A). However, counting the number of initiated colonies revealed a similar colony-initiating efficiency in this population, suggesting that these cells present colony initiation potential but failed to undergo proliferation (graph in Figure 4A).

### 3.5. Ex Vivo Progenitor Potential

To study CD274 expression in a more physiological model, we generated precision-cut lung slices (PCLSs) from one donor lung (patient 2, Figure 4A,B). After three days of culture, PCLSs were dissociated, and CD274 expression was analyzed in the epithelial compartment of three cultured PCLSs (Figure 4B). Cells isolated from the same patient at the time of PCLS generation were used as Day 0 control (Figure 4A). Flow cytometry analysis showed an overall increase in CD274 expression in both HTII-280^neg^ (Q3) and HTII-280^pos^ (Q2) populations, suggesting the expansion of CD274^pos^ cells in ex vivo conditions (Figure 4C). In the absence of lineage tracing, it is difficult to determine progenitor–progeny relationships in this assay. However, the increase in proportion in both populations strongly suggests the proliferation and differentiation of the original CD274^pos^ population, which is a hallmark of progenitor cells.

Interestingly, using mouse PCLSs to investigate the fate of lineage-traced alveolar epithelial cells, we also reported a massive loss of mature AT2s, as well as amplification of the IAAPs [11]. The increase in the percentage of HTII-280^pos^CD274^pos^ cells after 3 days of culture could, therefore, represent the differentiation of CD274^pos^ cells to HTII-280^pos^ cells.

Taken together, our data demonstrate the existence of a CD274^pos^ population in the human lung, which is expanded in the IPF lung, and which has a similar phenotype and functional behavior with the recently identified IAAPs in the mouse lung.

## 4. Discussion

The interplay between the immune system and structural cell types in the normal and fibrotic lung is a subject of high research interest. Several studies have thus far pointed to the role of the checkpoint molecule CD274 (PD-L1) in the mesenchymal compartment of IPF patients [32,33,34], while its role in the epithelial compartment remains to be established. We have previously shown that in the distal lung epithelium, CD274 expression is limited to an alveolar epithelial population with quiescent progenitor properties. In this study, we examined the cell-surface expression of CD274 in the epithelial compartment of the human donor and IPF lung. Mining the published single-cell NGS data (Kaminski/Rosas, Banovich, www.ipflcellatlas.org, (accessed on 15 December 2021)) of IPF lungs, we determined the *CD274* mRNA expression in AT2s, aberrant basaloid, and *KRT5-/KRT17+* epithelial cells. However, flow cytometry analysis of the functionally active, cell-surface-expressed CD274 identified a population with an intermediate AT2/conducting airway phenotype, which was significantly increased in the lung of patients with IPF. This population revealed similar molecular characteristics and in vitro behavior to the previously identified IAAPs [11]. Moreover, in an ex vivo model of donor PCLS culture, these cells were expanded, suggesting that they have self-renewal capacity. 

Although CD274 expression has been acknowledged in the alveolar and bronchiolar compartments of the IPF lung [22], this is the first study that assesses its cell-surface expression, which is related to its functional immune-suppressive role. In the context of IPF and bleomycin-induced lung fibrosis, CD274 induction plays a role in the SMAD3/GSK3-mediated fibroblast to myofibroblast fate induction and collagen deposition [33,34], and in the IPF fibroblasts’ invasive properties [32,35]. In our study, although we detected low levels of CD274 on the surface of mesenchymal cells (CD45^neg^ CD31^neg^ EpCAM^neg^), we did not find any significant change in expression between the donor and IPF single-cell preparations (data not shown). However, our cell isolation protocol used mild enzymatic dissociation (dispase), which was optimized for the extraction and enrichment of viable AT2 cells. In these conditions, mesenchymal cells are under-represented, and harsher enzymatic dissociation protocols are necessary for their extraction, in particular from IPF tissue [36]. Thus, in our publication, we refrain from concluding any difference in CD274 expression in the mesenchymal compartment of IPF patients. 

We and others have shown that in IPF, the expression of PD-L1’s cognate receptor PD-1 is increased on the surface of CD4+ T cells, strongly suggesting that the PD-1–PD-L1 (CD274) interaction plays an important functional role in the context of lung fibrosis (Appendix A and [33]). Indeed, PD-1-expressing CD4+ T cells secrete TGFβ and IL17A, which, in turn, activate fibroblast collagen 1 secretion via a STAT3-mediated mechanism [33]. In the mouse lung epithelium, expression of Cd274 was increased in a population marked by low expression of the Tomato AT2 lineage label (Tom^low^) [11], and this population actively replenished the mature AT2 pool following alveolar injury mediated by *Fgfr2b* deletion (termed injury-activated alveolar progenitors (IAAPs)) [12]. Similar to IAAPs, human donor CD274^pos^ cells did not expand significantly in alveolar-permissive 3D cell culture conditions. However, a higher number of CD274^pos^ AT2 and non-AT2 cells emerged in cultured donor PCLS. Further experiments are needed to establish lineage relationships between various populations of CD274-expressing cells in this model, but our data clearly show the expansion of this population. 

Expression of *CD274* has been noted at the mRNA level in AT2s in the previously published scNGS data (Kaminski and Banovich) (Figure 1), but the consistent cell-surface expression in this cell type was seen only after PCLS culture, suggesting different levels of regulation at the mRNA and cell-surface protein level. Interestingly, *CD274* mRNA expression was also noted in a very small subpopulation of AT2s (less than 0.45%), which expands in the lungs of chronic obstructive pulmonary dysplasia. This population has particular inflammatory properties, such as pro-inflammatory cytokine secretion, but CD274 expression at the cell-surface level in these cells has not been analyzed [23]. 

In this study, using formaldehyde-fixed IPF and donor lungs, we also performed IF with commercially available antibodies raised against a 17 amino acid peptide from near the center of human CD274 (data not shown). Despite optimization, we failed to detect a meaningful CD274 signal in these samples. In the future, it will be critical to determine the localization of CD274^pos^ epithelial progenitor cells in IPF and donor lungs using alternative methods.

The functional role of CD274 expression in the lung epithelium remains an open question. It is well known that CD274 plays an important immuno-suppressive role in various cancers, conferring immune evading capabilities to the expressing transformed cells [14,37]. However, CD274 is expressed widely in healthy tissues, suggesting that it has an important functional homeostatic role. Indeed, a variety of tissues and cell types (testis, brain, eye) are preferentially protected from the immune system, a phenomenon known as an immune privilege, a process in which CD274 plays an important role [38]. It also protects hematopoietic stem cells against premature removal from circulation [39] and marks a population of quiescent progenitors that regenerate Lgr5-expressing intestinal stem cells [40]. In the hair follicle, CD274 plays an important role in the progression of the hair cycle, potentially conferring protection to hair follicle stem cells located in the bulge area of the outer root sheath during the catagen phase [41]. Consistent with the role of CD274 in the immune tolerance of healthy tissues, treatment with anti-PD-1/PD-L1 antibody for various malignancies results in serious side effects such as gastrointestinal toxicity, skin rashes, and alopecia [42], as well as kidney, liver, and lung injury [43]. 

Checkpoint inhibitors (CPIs) are widely used in the treatment of non-small-cell lung adenocarcinoma (NSCLC). However, pneumonitis is an important side effect of CPI therapy (10–20% of patients) [44]. Moreover, the pre-existence of interstitial lung diseases before cancer diagnosis increases the risk of CPI-induced pneumonia, suggesting an important functional role of the population of PD-L1-expressing cells [16]. Our results, although not fully proven, point to the direction of an immune-protective role of a potential epithelial progenitor in the human lung, whose precise identity and lineage composition remains to be further elucidated. Several studies proposed that CPI therapy might be beneficial to limit the fibro-proliferative reaction in IPF [32,35].

Indeed, CD274 expression downstream of P53 was reported to be increased in IPF vs. donor human lung fibroblasts and correlates with the invasive capabilities of IPF fibroblasts in vitro. In addition, blocking CD274 using neutralizing antibodies following injection of CD274^pos^ IPF fibroblast in SCID mice, a humanized mouse model, attenuated fibrosis development [32]. It was also shown that CD274 expression downstream of TGFβ1 mediates lung fibroblasts to myofibroblast transition through SMAD3 and GSK3b/β-catenin signaling [34,37]. However, our data, together with the clinical studies mentioned above, suggest that CD274 inhibition, unless specifically targeted, might further injure the already precarious lung epithelial compartment in IPF. 

## Figures and Tables

**Figure 1 cells-11-01593-f001:**
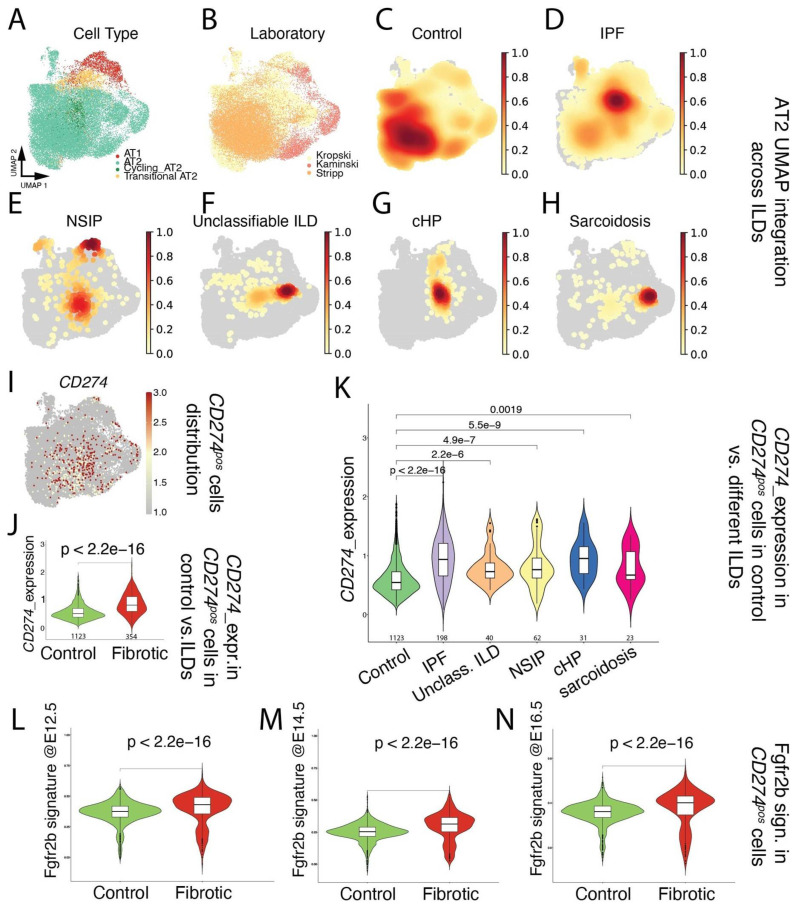
Single-cell transcriptome analysis of distal epithelium in control and end-stage fibrotic donors: (**A**) dimensional reduction in integrated data, showing cell-type distribution, visualized by UMAP; (**B**) dimensional reduction in integrated data derived from 3 published datasets, visualized by UMAP, with cells colored by lab of origin; (**C**–**H**) distribution of cell density by disease. IPF: idiopathic pulmonary fibrosis, NSIP: non-specific interstitial pneumonia, unclassifiable ILD: interstitial lung disease, cHP: chronic hypersensitivity pneumonitis; (**I**) expression of *CD274* (*PD-L1*), visualized by UMAP; (**J**) expression of *CD274* (*PD-L1*) in positive cells, comparing control and fibrotic datasets, visualized by violin plots. The value underneath each violin represents number of expressing cells; (**K**) expression of *CD274* (*PD-L1*) in positive cells, comparing control and each fibrotic disease, visualized by violin plots. The value underneath each violin represents number of expressing cells; (**L**–**N**) expression of mouse Fgfr2b signatures identified at E12.5, E14.5, and E16.5 on human *CD274* (*PD-L1*)-positive cells, comparing control and fibrotic datasets, visualized by violin plots.

**Figure 2 cells-11-01593-f002:**
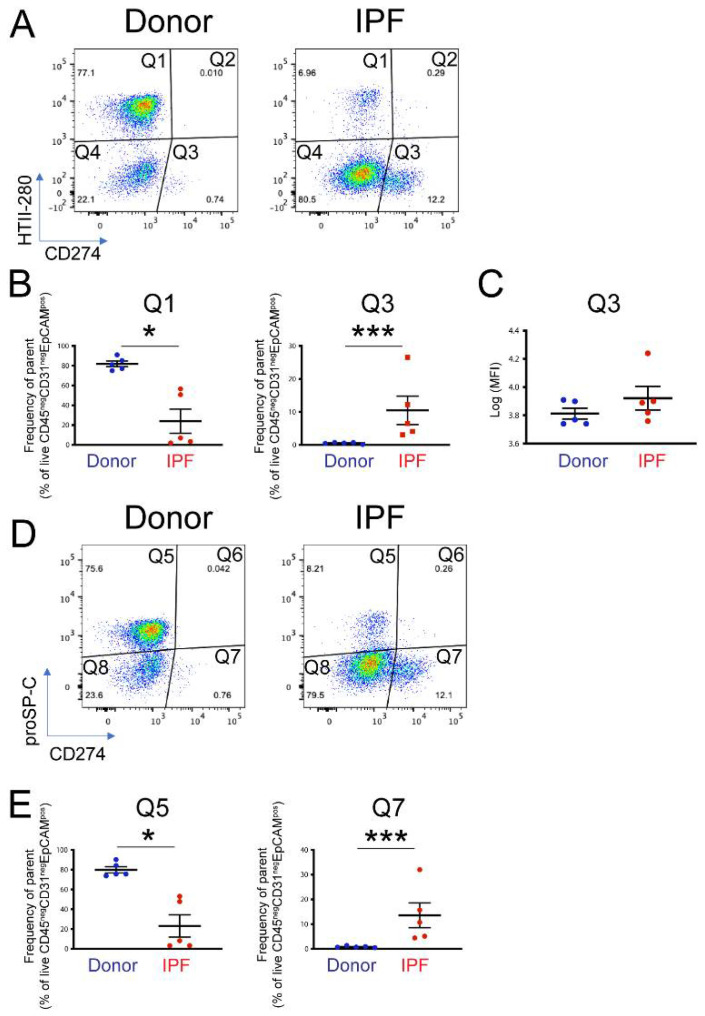
Cell-surface expression of CD274 in donor and IPF human lungs: (**A**) representative flow cytometry panels of HTII-280 vs. CD274 expression in the epithelial compartment (CD45^neg^ CD3^neg^ EpCAM^pos^) of donor and IPF lungs; (**B**) quantification of the HTII-280 vs. CD274 data shown in (**A**); (**C**) quantification of the mean fluorescence intensity (MFI) of the CD274^pos^ (Q3 population) in donor and IPF samples; (**D**) representative flow cytometry panels of proSP-C vs. CD274 expression in the epithelial compartment (CD45^neg^ CD31^neg^ EpCAM^pos^) of donor and IPF lungs; (**E**) quantification of the proSP-C vs. CD274 data shown in (**D**). Statistical analysis was performed on log-transformed values, and Student’s *t*-test was applied to determine statistical significance. * *p* < 0.05, *** *p* < 0.001.

**Figure 3 cells-11-01593-f003:**
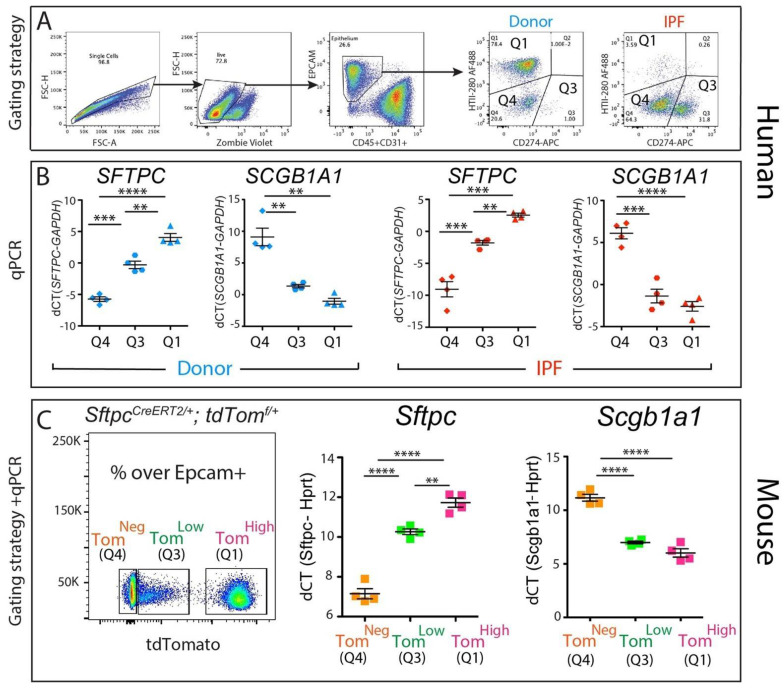
Molecular phenotype of the CD274^pos^ population: (**A**) representative sorting strategy to isolate HTII-280^pos^ CD274^neg^ cells (Q1), HTII-280^neg^ CD274^pos^ cells (Q3), and HTII-280^neg^ CD274^neg^ cells (Q4) in donors (n = 4) and IPF (n = 4); (**B**) corresponding *SFTPC* and *SCGB1A1* mRNA quantification by qPCR in Q1, Q3, and Q4 subpopulations in donors (n = 4) and IPF (n = 4); (**C**) representative flow cytometry panel of tdTomato expression in the epithelial cell compartment of tamoxifen-treated *Sftpc^CreERT2/+^*: *tdTomato^flox/flox^* mice (left panel). qPCR analysis of *Sftpc* and *Scgb1a1* in the populations sorted according to the level of tdTomato expression (middle and right panel). Data are represented as means ± SEM. ** *p* < 0.01, *** *p* < 0.001, **** *p* < 0.0001.

**Figure 4 cells-11-01593-f004:**
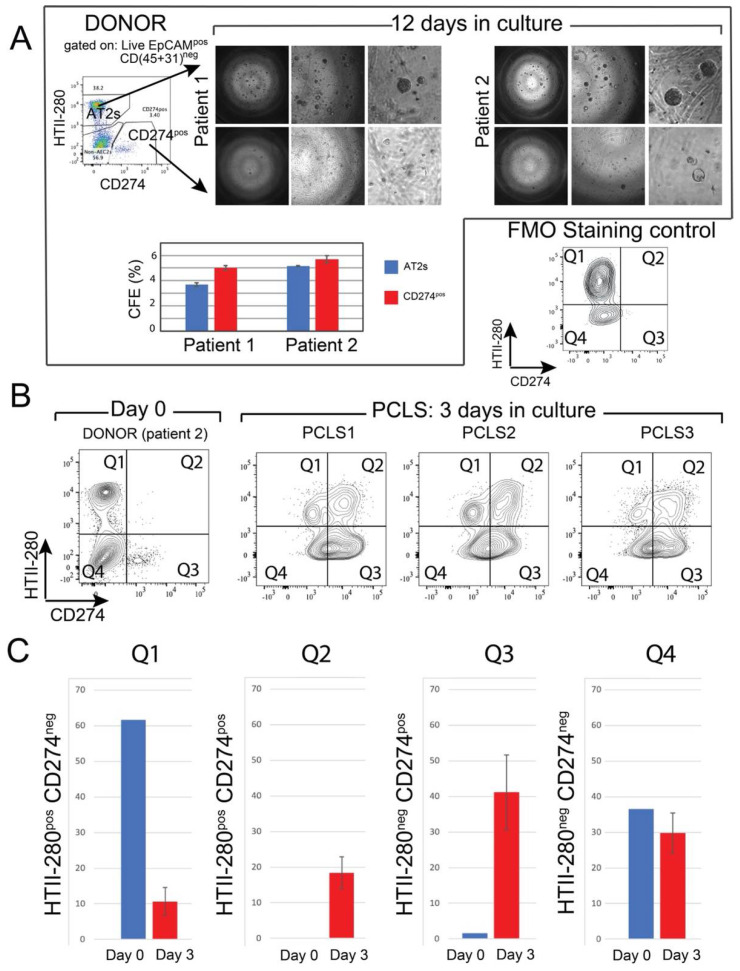
In vitro clonogenic potential of human CD274^pos^ epithelial cells: (**A**) representative images of AT2- and CD274^pos^-derived organoids from two donor patients. Quantification of the CFE; (**B**) flow cytometry panel of the day 0 analysis of HTII-280/CD274 expression in the epithelial compartment of patient 2 used for PCLS generation and culture. Flow cytometry analysis of HTII-280/CD274 expression in the epithelial compartment of PCLSs (n = 3) cultured for 3 days compared with the cells isolated at day 0 from the lung of patient 2; (**C**) quantification of the data in (**B**).

## Data Availability

Transcriptomic data are publicly available at the following location: Microarray data for the isolated AT2s and PD-L1^pos^ cells from donor and IPF lungs are available under the accession number GSE195770.

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
