# Peer review of "Cell-Surface Programmed Death Ligand-1 Expression Identifies a Sub-Population of Distal Epithelial Cells Enriched in Idiopathic Pulmonary Fibrosis"

_cells, 2022, doi:10.3390/cells11101593_

Round 1

Reviewer 1 Report

In this paper Ahmadvand et al analyzed the available RNA-seq dataset in the IPF cell atlas to investigate the presence of CD274 positive IAAP  cells in human lungs. Afterwards the authors quantified by flow cytometry the abundance of IAAP equivalent cells, as well as AT2 equivalent cells in IPF vs. donor human lungs. Moreover they investigated the expression of key genes by qPCR and monitored the growth of these cells in organoid and PCLS assays. The data they show indicate that IAAPs are also present in humans, and are differentially regulated in IPF  vs. donor lungs. That is of high interest because CD274 expression of lung epithelium provides a possible link between immune cells and structural cells.

This is a very nice well performed study! It is well written performed with state of the art methods. There is nothing to add from my side. I would recommend to accept it immadiately.

Congratulation to this well performed study!

Author Response

Answer to Reviewer 1: We would like to thank this reviewer for this positive evaluation. We are also excited about the possible link between IAAPs and immune cells via PD-L1.

Reviewer 2 Report

In this manuscript, the authors investigated the presence of equivalent IAAP PD-L1 positive cells in human lungs by use for the available scRNA-seq dataset in the IPF cell atlas, the abundance of IAAPs and mature AT2s, the expression of key genes as well as the growth of these cells in organoid and precision-cut lung slice assays in IPF vs. donor human lungs. Based on the multiple approach, the authors demonstrated the existence of a PD-L1-positive population in the normal human lung, which is expanded in the IPF lung similar with the IAAPs observed in the mouse lung regarding the phenotype and functional behavior. Although this study provides valuable clinical data, the reviewer suggests some serious concerns as follows.

Major Comments

  1. In the Introduction, the authors should add enough explanation regarding Sftpc, Ccnd1/2, Etv5, Fgfr, etc, which helps readers to understand the background of this manuscript.
  2. Could the authors also assess the cell-surface expression of PD-1 related to its functional immune-suppressive role, in order to further seek out the novelty of the present study?
  3. It is highly recommended to measure the cell-surface expression level of PD-L1 in the mesenchymal cells by use of the strong enzymatic dissociation reagent to determine whether any difference between IPF patients and donor are observed.
  4. In the Discussion or Introduction, the authors should describe the role of PD-L1 in the SMAD3/GSK3-mediated myofibroblast fate induction and collagen deposition as well as the invasive property of fibroblasts derived from IPF patients more detail, because it seems to be hard to understand the mechanism in the present form.

Minor Comments

  1. Throughout this manuscript, in-text citation style should be revised correctly.
  2. In Line 247, “focussing” and “mined” seem to be typos.

Author Response

Major Comments

  1. In the Introduction, the authors should add enough explanation regarding Sftpc, Ccnd1/2, Etv5, Fgfr, etc, which helps readers to understand the background of this manuscript.

Following pneumonectomy, Tomlow AT2s are activated and display progenitor-like properties as they are amplified and exhibit elevated expression of Fgf signaling genes Fgfr2b, Etv5, AT2 differentiation marker Sftpc, and cell cycle genes Ccnd1, Ccnd2 expression compared to sham.

These changes are consistent with the activation of the progenitor like properties of the  IAAPs which would allow them to proliferate and differentiate towards mature AT2

  1. Could the authors also assess the cell-surface expression of PD-1 related to its functional immune-suppressive role, in order to further seek out the novelty of the present study?

This is indeed a very important point that we will develop in further studies involving the use of both mouse and human IAAPs. The current manuscript just aims to establish the presence of human IAAPs, which is already a significant step in our understanding of the AT2 lineage in human.

  1. It is highly recommended to measure the cell-surface expression level of PD-L1 in the mesenchymal cells by use of the strong enzymatic dissociation reagent to determine whether any difference between IPF patients and donor are observed.

We agree that this is also important but as we focused on the epithelial cells (our digestion protocol did not allow us to capture high quality mesenchymal cells), this would mean redoing the whole study from scratch. Given the time-sensitive nature of the special issue, this work will have to be done in a separate study.

  1. In the Discussion or Introduction, the authors should describe the role of PD-L1 in the SMAD3/GSK3-mediated myofibroblast fate induction and collagen deposition as well as the invasive property of fibroblasts derived from IPF patients more detail, because it seems to be hard to understand the mechanism in the present form.

 We have now introduced in the discussion a paragraph dealing with PD-L1 expression in the myofibroblasts

Indeed, CD274 expression, downstream of P53, was reported to be increased in IPF vs Donor human lung fibroblasts and correlates with the invasive capabilities of IPF fibroblasts in vitro. In addition, blocking CD274 using neutralizing antibodies following injection of CD274pos IPF fibroblast in SCID mice, a humanized mouse model, attenuated fibrosis development [33]. It was also shown that CD274, downstream of TGFb1, mediates lung fibroblasts to myofibroblast transition through SMAD3 and GSK3b/b-catenin signaling [35,,39]. However, our data, together with the clinical studies mentioned above, suggest that CD274 inhibition, unless specifically targeted, might further injure the already precarious lung epithelial compartment in IPF.

Minor Comments

  1. Throughout this manuscript, in-text citation style should be revised correctly.

We have edited the references in the text.

Reviewer 3 Report

This is a very well designed and written paper. It provides very important data in the IPF field as it demonstrates the normal human lungs possess injury activated alveolar progenitors that are enriched in the IPF human lung. My only concern is that the organoids have been performed with Matrigel, a mouse sarcoma ECM mixture, and thus I would like the authors to make a comment on the use of more relevant ECM matrices. Other than that I see no flaw in the study. The statements made in the last paragraph of the Discussion are to be highlighted. Great effort to all the authors.    

Author Response

Answer to Reviewer 3: We would like to thank this reviewer for this positive evaluation.

Comment on the use of more relevant ECM matrices: We have used Matrigel® growth factor-reduced (Corning). This ECM mix is commonly used to generate organoids. However, other synthetic matrices could also be used in the future.

Reviewer 4 Report

This manuscript shows results of high potential originality. Nevertheless, I ask the Authors only a little bit effort to better explain the significance and the limits of their results.

Furthermore, for the common readers some statements are lacking of a comprehensive bibliography. For example, is not referenced the statement that lineage tracing experiments on CD274 cells show that they belong to the AT2 cell population. Also the fact that CD274 cells have a self-renewal capacity should be referenced.

The manuscript has a conclusion about the checkpoint inhibitors (CPI) drugs, whilst the abstract has not. Please, conform the abstract conclusion with the article one or viceversa.

Author Response

This manuscript shows results of high potential originality. Nevertheless, I ask the Authors only a little bit effort to better explain the significance and the limits of their results.

Critique 1: Furthermore, for the common readers some statements are lacking of a comprehensive bibliography.

Answer: There is fortunately for us no comprehensive bibliography as we are the first one describing the existence of the Pdl1Pos IAAPs (Ahmadvand, N.; Khosravi, F.; Lingampally, A.; Wasnick, R.; Vazquez-Armendariz, I.; Carraro, G.; Heiner, M.; Rivetti, S.; Lv, Y.; Wilhelm, J.; et al. Identification of a Novel Subset of Alveolar Type 2 Cells Enriched in PD-L1 and Expanded Following Pneumonectomy. The European respiratory journal 2021, 58, doi:10.1183/13993003.04168-2020.)

Considering this, this is what we wrote in the abstract

we previously identified a lung population of quiescent injury activated alveolar epithelial progenitors (IAAPs) marked by low expression of the AT2 lineage trace marker tdTomato (Tomlow), and characterized by high levels of Pd-l1 (Cd274) expression

And in the introduction

Using SftpcCreERT2/+: tdTomatoflox/flox mice, and based on the differential level of the tomato reporter between two lineage-traced alveolar epithelial subpopulations, we have previously reported the existence of a novel population of AT2s, named Tomlow AT2s and which are enriched in programmed death-ligand1 (Pd-l1) [11]. This AT2 subpopulation expresses low levels of Sftpc, Sftpb, and Sftpa1 and displays a low level of Fgf signaling activation. Following pneumonectomy, Tomlow AT2s are activated and display progenitor-like properties as they are amplified and exhibit elevated expression of Fgf signaling genes Fgfr2b, Etv5, AT2 differentiation marker Sftpc, and cell cycle genes Ccnd1, Ccnd2 expression compared to sham. These changes are therefore consistent with the activation of the progenitor-like properties of the IAAPs which would allow them to proliferate and differentiate towards mature AT2s

The main reference for this statement is

Ahmadvand, N.; Khosravi, F.; Lingampally, A.; Wasnick, R.; Vazquez-Armendariz, I.; Carraro, G.; Heiner, M.; Rivetti, S.; Lv, Y.; Wilhelm, J.; et al. Identification of a Novel Subset of Alveolar Type 2 Cells Enriched in PD-L1 and Expanded Following Pneumonectomy. The European respiratory journal 2021, 58, doi:10.1183/13993003.04168-2020.

Critique 2: For example, is not referenced the statement that lineage tracing experiments on CD274 cells show that they belong to the AT2 cell population. Also the fact that CD274 cells have a self-renewal capacity should be referenced.

Lineage tracing was done on the AT2s using the SftpcCreERT2/+: tdTomatoflox/flox mice. Based on the differential level of the tomato reporter between two lineage-traced alveolar epithelial subpopulations, we reported the existence of a novel population of AT2s, named Tomlow AT2s and which are enriched in programmed death-ligand1 (Pd-l1) [11]

We did not do the lineage tracing of Cd274 cells.

Critique 3: The manuscript has a conclusion about the checkpoint inhibitors (CPI) drugs, whilst the abstract has not. Please, conform the abstract conclusion with the article one or viceversa.

We have added this conclusion to the abstract:

CD274 function in these cells as a checkpoint inhibitor may be crucial for their progenitor function suggesting that CD274 inhibition, unless specifically targeted, might further injure the already precarious lung epithelial compartment in IPF.

Round 2

Reviewer 2 Report

Authors have addressed most of the concerns from Reviewer on revised manuscript. The present paper seems to be accepted for publication in Cells.